# The Perceived Stigma Reduction Expressed by Young Adults in Response to Suicide Prevention Videos

**DOI:** 10.3390/ijerph18126180

**Published:** 2021-06-08

**Authors:** Sarah Keller, Vanessa McNeill, Tan Tran

**Affiliations:** 1Department of Communication, Montana State University Billings, Billings, MT 59102, USA; 2Department of Psychology, Montana State University Billings, Billings, MT 59102, USA; vanessa176@hotmail.com; 3Statistical and Consulting Research Services, Montana State University, Bozeman, MT 59718, USA; vinhtantran@gmail.com

**Keywords:** stigma, help-seeking, suicide, suicidal ideation, tests of significance, practical meaningfulness, targeting and tailoring public health messages, mixed methods

## Abstract

Evidence indicates that stigma impedes an individual’s chance of seeking professional help for a mental health crisis. Commonly reported aggregate-level results for stigma-reduction efforts obfuscate how much stigma reduction is needed to incur a practically meaningful change within an individual, defined here as an attitudinal shift and openness towards seeking mental health for oneself and/or support for others. When basing conclusions and recommendations about stigma-reducing interventions on aggregate scales, it is unclear how much stigma reduction is needed to incur meaningful change within an individual. We explored the impact of reductions in stigma of help-seeking scores in response to an online suicide prevention video among young adults in the United States, using online surveys to collect qualitative and quantitative data. We compared mean changes in the stigma scores from pre- to post-test (video exposure) of 371 young U.S. adults using standard *t*-tests and individual level analysis. A separate thematic analysis of free-text responses was also conducted from a smaller, randomly-selected subgroup, capturing individuals’ attitudes towards help-seeking for mental health problems. Great attention was given to participants to ensure that they were in a campus setting where counseling services were available. Four main themes emerged: (1) small changes in stigma scores were associated with individual reports of meaningful reductions in their attitudes towards professional counseling; (2) increased empathy towards victims of suicide and other mental health problems sometimes indicated increased empathy for victims of suicide and decreased openness in professional help; (3) empathy towards victims sometimes took the form of increased scores and grief or sadness, possibly thwarting the potential for help-seeking; and (4) self-reports of decreased stigma were not always associated with increased openness towards help-seeking. Results also indicated that small stigma score charges, not meeting statistical significance, were often associated with increased openness towards seeking help. These findings, discovered using mixed-methods, contribute to the body of literature regarding stigma towards suicide and help-seeking by demonstrating deficits in the aggregate-only analysis of stigma-reducing interventions specifically aimed at suicide prevention. Such individuation in stigma experiences indicates that public education on how to reduce the stigma of help-seeking for suicide prevention needs to consider individual-level analyses for improving target populations. Recommendations for future research include additional studies prior to releasing suicide prevention videos to public forums where they may be seen by individuals without access to help.

## 1. Introduction

According to the Substance Abuse and Mental Health Services Administration (SAMHSA), in 2019, 12 million adults had serious thoughts of suicide; 3.5 million adults made suicide plans; and 1.4 million attempted suicide [1]. The 2019 U.S. National Action Alliance for Suicide Prevention’s goal is to reduce annual suicide rates by 20% by 2025 [2]. Suicide has surpassed car accidents as the number one cause of injury-related death in the United States. There has been a 30% increase in the number of suicides in the United States since 1998 [3].

In the focused age-group of this study, young (18–25) college-age adults, suicide is now the second-leading cause of death [4]. Most attempts originate with suicidal ideation [1], escalating from ideation to an attempt without help from professionals [5]. Currently in Montana, where the authors teach and conducted two of the three studies detailed herein, the suicide rates are the highest in the United States [6] and climbed from 23.9 per 100,000 in 2014 to 28.9 per 100,000 [7] in 2016.

Many factors contribute to suicidal ideation; the most common contributors are social isolation and stigma [8,9,10,11]. Evidence indicates that stigma appreciably impedes an individual’s chance of seeking professional help for suicidal thoughts [12,13,14,15] “over and above the impairments related to the mental disorders themselves” [16]. In fact, national and regional disparities in suicide rates are associated with differences in stigma [17]. Reynders et al. [18] found that people in the Netherlands, where suicide rates are low, have more positive attitudes toward help-seeking and experience less self-stigma and shame compared to samples where suicide rates are relatively high.

Stigma is generally defined as a set of individual attitudes and beliefs substantiating a perception that one is discredited by society and condemned to an undesirable social status, minimalizing the individual “from a whole and usual person to a tainted, discounted one” [18] (p. 3). Stigma may also be defined as a process involving labelling, separation, stereotype awareness, stereotype endorsement, prejudice, and discrimination in a context in which social, economic, or political power is exercised to the detriment of members of a social group [19]. People who perceive themselves as outcasts (thwarted belongingness) and/or a burden to society (perceived burdensomeness) may feel more socially isolated and less likely to seek help [5]. In relation to suicide, researchers have identified at least three main categories of stigma: stigma of suicide, stigma of seeking help, and self-stigma.

### 1.1. Stigma of Suicide, Help-Seeking, and Self

Suicide stigma and stigma of professional services can interfere with help-seeking by reducing an individual’s perceived need for help [17], impairing adherence to intervention [20], decreasing self-esteem [21], and increasing social isolation [22,23]. Reynders et al. found that stigma predicted both professional and informal help-seeking behavior [24]. The role of stigma in help-seeking for suicide appears to be two-directional; it both acts as a barrier and is more prevalent among those most at risk. Kearns et al. [25] found that stigma among university students was a key barrier to accessing university professional services for suicide risk. Meanwhile, an online survey in Australia by Calear, Batterham, and Christensen [26] indicated that respondents experiencing suicidal ideation had more negative attitudes toward help-seeking and lower intentions to seek help compared to those without suicidal thoughts. One meta-analysis conducted by Clement et al. [27] identified 56 studies (with a total of 26,313 participants) reporting the association between stigma and help-seeking for mental health problems, including suicide. A majority of the studies [28,29,30] reported a statistically significant association between stigma and help-seeking, with the effect sizes calculated from the reported statistics ranging from Cohen’s d = −2.73 to 0.36 (a median of −0.27, with negative values indicating that stigma reduces help-seeking).

Related to stigma is a common feeling of anxiety about death that may impede forthcoming communication about one’s feelings about life and death, possibly reducing the open disclosure of thoughts about suicide. Stizzi et al. (2018) found that death anxiety could be reduced by educating young people about the topic and using psycho-drama exercises to role play different thoughts and experiences around the topic [25].

Self-stigma may also obstruct a person’s willingness to take any action on their own behalf when confronted with a health threat when they perceive the threat to be stigmatizing [27,28]. Goffman referred to self-stigmatized individuals as “wise” [20] (p. 4), that is, they knew they were stigmatized and therefore withdraw from society (isolate). Chen et al. [30] recently demonstrated that once the psychological distress of self-stigma is removed, an individual’s social isolation becomes moot. Unfortunately, few studies have attempted to provide information regarding how much stigma reduction is needed to reduce the risk of suicide or increase the chance of help-seeking. The Self-Stigma of Help-Seeking scale was designed to measure a person’s attitudes towards professional psychological help-seeking, what they perceive as others’ views of professional counseling, and how they view others who seek professional psychological help [29]. The scale uses an agree–disagree scale on 12 or 6 points to measure perceived public stigma, willingness to seek psychological help, attitudes towards seeking professional help, and self-stigma. In the proposed model, one’s self-stigma and attitudes towards seeking help mediate the relationship between perceived public stigma and willingness to seek help [29].

### 1.2. Pitfalls of Aggregate Measures

Clement et al. [27] identified 144 studies with 90,189 participants examining the role of mental health-related stigma in help-seeking for mental health problems. Though the review identified a median association between stigma and help-seeking at d = −0.27, all of the studies used mean effect sizes to determine the impact of stigma, obfuscating the variations among individuals. Similarly, Schomerus et al. [17] used country-level data to determine that national variations in suicide rates are correlated with variations in cultural beliefs about mental illness and stigma. Further, literature showing that reductions in the mean levels of stigma of help-seeking (measured by the Self-Stigma of Seeking Help scale by Vogel, Wade and Haake, 2006) are tied to mean increases in help-seeking behavior was also analyzed at the group level. Vogel, Wade, and Haake found that mean differences of 3.0 in SSOSH score (M = 24.3 and SD = 6.1 vs. M = 27.3 and SD = 6.6) predicted whether or not college students would seek psychological services over a two-month period. Unfortunately, Vogel et al. did not report how many individuals displayed reductions in stigma scores, nor did they report characteristics or qualitative data associated with those individuals. Commonly reported aggregate-level results for stigma-reduction efforts obfuscate how much stigma reduction is needed to incur a practically meaningful change within an individual, defined here as an attitudinal shift increasing the likelihood that a person will seek professional help for suicidal thoughts. All of the association studies we could find used summary statistics to investigate the impact of stigma at the aggregate level and did not provide inferences about the potential impact of stigma occurring for individuals at risk of suicide. Schomerus et al. [17] used country-level data to investigate national variations in suicide and stigma. Some researchers have argued that important nuances within a rich dataset should be more thoroughly analyzed [14,16]; in any dataset, it is possible only a few individuals experience large changes while most individuals have no change or change in the opposite direction.

### 1.3. Video as Intervention for Stigma Reduction

This study examined young U.S. adults’ perceived stigma reductions after viewing suicide prevention videos, followed by comparing individual comments with their quantitative scores. One goal of this study was to assess whether online videos featuring stories of young adults’ experiences with depression and suicide could deliver a strong enough message to young adults (peer-to-peer) to prompt enough of an attitude change in stigma to increase the likelihood of help-seeking behavior in the future. Another goal was to use a theoretically driven, person-based research approach to intervention development, using suicide prevention videos as our stimuli. Research by Aseltine and DeMartino [31] indicated that the *SOS* curriculum was associated with reduced suicidal attempts of students in school settings. Research by Keller and Wilkinson [26] showed the *Let’s Talk* live theatre performance (on which the *Let’s Talk* video documentaries were based) increased students’ self-efficacy and perceived response-efficacy for help-seeking. Qualitative research by Keller, Austin, and McNeill [31] indicated that students involved in the *Let’s Talk* plays, as either audience members or actor/actresses, reported reduced isolation and a subsequent increase in willingness to seek help in a crisis. With so many young adults interacting with video-content through forums and social media, we explored using video-content as a potential avenue for suicide prevention messaging, targeted at young adults. Our prior research [32] indicates that any decrease in stigma of help-seeking can increase an individual’s chance of accessing professional help, a known preventive measure for suicide [33]. However, no literature is available to indicate how much stigma reduction is needed for an individual to experience change [34] and existing measures of stigma and other attitudinal variables are imperfect [31,32]. Given this imperfection, the available measurement tools are likely to capture changes in stigma that may or may not be meaningful. Knowing the significant impact of the *SOS* curriculum and the *Let’s Talk* live performances, in regards to lowering stigma towards help-seeking, we hypothesized that the *Let’s Talk* documentaries, presented to young adults online could produce comparable results. Second, we wanted to further explore our prior results demonstrating that even small stigma changes not considered statistically significant, according to standard tests of significance, still warrant consideration. *Let’s Talk* videos are accessible to anyone at www.letstalkbillings.com (accessed on 8 November 2020). Access to the *SOS* video is part of a paid curriculum. Special access was provided for this project by the *SOS* Project Team.

## 2. Materials and Methods

The present study is concerned with examining the role of stigma reduction interventions at the individual level. Our outcome variable of interest is an attitude shift towards help-seeking behavior from formal services, specifically professional counseling (primary care or secondary/tertiary mental health services), including talking therapy services. The term ‘help-seeking’ is used to denote all stages of the process, encompassing initiation of, and engagement with, care.

### 2.1. Study Design

In order to determine how much change in perceived stigma is considered *meaningful* at the individual level, insofar as enough of a shift in attitude that leads to an increased likelihood of seeking help, this study examines three pilot studies; all designed to test a 13-min *Let’s Talk* documentary and 7-min *Let’s Talk* documentary with an evidence-based suicide prevention video, *Signs of Suicide (SOS)* 17-min video [31,32]. The impact of these three suicide prevention videos on measures of young adults’ stigma of help-seeking were examined, starting with a range of evidence-based scales used in prior suicide prevention research [14]. The Self-Stigma of Seeking Help scale (SSOSH) by Vogel, Wade, and Haake [29] was used as the basis for our aggregate analysis based on our pilot results coupled with literature showing that reductions in estimated mean perceived self-stigma are tied to increases in help-seeking behavior [24,35,36,37,38,39,40]. In Vogel’s groundbreaking paper based on a sample of 655 college students, they found that there was a difference of 3.0 (*p* = 0.032 from an unequal variances *t*-test) in the estimated mean SSOSH scores between a group of students who sought psychological services over a two-month period and those who did not. We used this difference of 3.0 as our cut-off measure for identifying meaningful change.

We used the Extended Parallel Process Model (EPPM) [41,42] to examine the threat and efficacy components related to the goals of the *Let’s Talk* program [14]. The EPPM was used as our theoretically driven, person-based qualitative research approach. The EPPM theoretical framework has been used effectively to both generate and evaluate messages intended to motivate health-related behaviors. The EPPM posits that respondents are more likely to change their behavior in response to a health message if both the perceived threat and their perceived efficacy are high. The EPPM also warns that videos that increase perceived threat without simultaneously increasing perceived efficacy to make a change can stimulate unhealthy coping mechanisms and increase suicidal ideation.

#### 2.1.1. Studies 1, 2, and 3

To examine the effectiveness of the *Let’s Talk* documentaries about the importance of seeking professional help for depression and thoughts of suicide, self-administered questionnaires were privately disseminated online. Lower rates of self-stigma of seeking help (SSOSH) were observed among students in pre-pilot studies using the 7-min *Let’s Talk* video—a documentary based on the live play “Under the Blanket” performed by Montana State University Billings college students.

Two additional pilots were conducted to further test the more effective intervention video on SSOSH. While mean SSOSH score decreases continued to be greater in the intervention group(s) compared to controls, methodological flaws in our pilots prevented us from comparing individuals at baseline to individuals at follow-up. Nevertheless, there’s enough evidence from all of the studies to warrant further research.

Before Studies 1, 2, or 3, initial small-scale pre-pilot studies demonstrated that the 7-min *Let’s Talk* suicide prevention video had statistically significant preliminary positive results for reducing stigma and increasing attitudes towards help-seeking.

In Study 1, the first part of the study involved an online experiment to examine the effectiveness of two suicide prevention videos—a one shorter video and one longer video from the community-based, narrative prevention program (*Let’s Talk*) (7 and 13 min long). The survey was administered online as a 3-armed randomized trial (two intervention groups and one waitlisted control group). Following the quantitative component of the survey (fixed-response questions), 10 open questions elicited free-text data. These probed specific domains related to stigma likely to be impacted by exposure to the video and were worded to be neutral and non-leading:“In the next 12 months if you were to experience a mental illness, how likely are you to seek help?”“How did the video make you feel?”“Did the video change your attitude towards mental health problems? If so, how?”“Did the video influence your attitudes towards professional counseling? If so, how?”“What, if anything, did you like about the video?”“What, if anything, did you dislike about the video?”“What would you change or add to the video, to make it more effective with people like yourself?”“How did you feel about the actors/actresses in the video?”“What would you say was the main point or message of the video?”“Do you think people who are suicidal are treated fairly in your community? Why or why not?”“Would a video like this help people become more accepting of suicidal risk? If so, how? If not, why not?”

Responses had no upper word limit, with instructions to provide as much or little detail as wished and to skip any questions that did not apply.

Study 1 A, in a second round, the same survey and videos were issued at a sister-campus (Montana State University Bozeman) to test the video on additional recruitments of college students.

In Study 2, the 7-min *Let’s Talk* video was used as one intervention and the *SOS* video was used as a second treatment in a 3-armed randomized trial (two intervention groups and one waitlisted control group). We used a national online panel, monetary incentives, and closed surveys administered at baseline and two weeks post-test to collect quantitative and qualitative data from a sample of college-enrolled individuals aged 18–25. We focused on young adults, as this was the suicide risk group targeted by the video interventions we were hoping to assess and because if enrolled in college, this group would have some access to mental health services. All individuals were anonymously contacted through Qualtrics Survey Software, with an email consent form inviting participants to click on a link to the survey if they wished to proceed. No personally identifying data were ever available to the researchers. Recruitment continued until each arm had an equal number of males to females and was large enough to achieve our goal of 100 participants per arm after the anticipated attrition. By using a third-party administrator to recruit and re-contact participants, we were able to protect the identities of participants and obtain data that were anonymous prior to analysis. Monetary incentives were calculated at minimum wage, $8.15/h, pro-rated for the estimated amount of time spent. Since two surveys, believed to last at least 20 min each, were combined with videos lasting from 7 to 17 min long, participants were anticipated to spend up to one hour on the study. Therefore, each participant was given $8.15 to participate, awarded upon the completion of the follow-up survey.

In Study 3, the survey was administered as an online pre/post one-group design. The shorter *Let’s Talk* video was used alone. The same quantitative survey was used, along with a pared-down list of qualitative questions as follows: (1) how did the video make you feel, (2) did the video change your attitude towards mental health problems, and (3) did the video change your attitude towards professional counseling? These open-ended questions were posed immediately after viewing the video in the baseline survey and at one-week post-test (Figure 1A–C).

All of the above studies’ protocols were approved by the Montana State University Billings Institutional Review Board (IRB00001622).

#### 2.1.2. Data Collection

Closed surveys were administered through the Qualtrics survey platform before showing a video to participants. Quantitative and qualitative data were collected immediately after the video and again either one-week or two-weeks post-intervention from intervention groups and controls. All self-identifying information was removed to protect the students’ anonymity prior to analysis by the research team. No personally identifying data were ever available to the researchers. Potential participants received an email briefly describing the purpose of the study, a consent form inviting them to click on a link to the survey if they wished to proceed in exchange for extra-credit (or monetary rewards for the paid, national sample), and suicide prevention resources. Though we focused on young adults aged 18–25, some of the individuals in the studies were non-traditional students and significantly older (ranging in age from 25 to 75). No penalty was tied to participation, and students were made aware that one of their professors was a researcher in the study. Two of these studies were offered to Montana State University Billings (MSUB) students attending online Introduction to Psychology courses (Studies 1 and 3).

All participants were given suicide prevention resources after each data collection, as well as invitations to view any of the intervention videos. Students not interested in participating were offered an alternative project worth equal value.

### 2.2. Online Experiment

All participants were invited to take part in a survey evaluating a “Digital Stigma-Reduction Intervention”, with the survey link provided in the sampling email. All participants provided online informed consent and were provided with a list of support sources. Three online studies were conducted to assess the potential impact of short, targeted videos on the stigma of help-seeking—a key barrier to effective suicide prevention messaging. Subjects were randomly assigned to one of three conditions: two video treatment groups and one control group.

The *Let’s Talk* treatment used comprises documentary-style videos of theater-student volunteers who, under the guidance of a trained mental health professional and a theater director, wrote and performed a theatre production to promote open dialogue about suicide risk and help-seeking for their campus peers and community-members.

We analyzed mean changes in quantitative self-reports of change as measured by the SSOSH scale between groups (Vogel, Haake, and Wade, 2006). The SSOSH scale consists of 5-point Likert items coded from 1 = strongly disagree to 5 = strongly agree. SSOSH has items that are summed to produce a score from 10 to 50.

#### 2.2.1. Procedure: Experiment

All three studies involved an online experiment to examine the effectiveness of suicide prevention videos. The survey questionnaire included the SSOSH scale, along with demographic questions and additional scales to capture participants’ mood states, perceptions of narrative performance, and perceived stress. Only the SSOSH scores were used for this analysis to determine how much score change in stigma was associated with self-reports of meaningful experiences of changing attitudes towards help-seeking in response to open-ended questions. Open-ended questions were included in the post-test to elicit qualitative responses.

#### 2.2.2. Data Analysis: Quantitative

The mean changes in SSOSH between baseline and follow-up across the three studies were measured, while controlling for age, gender, sexual orientation, and ethnicity, with a linear regression model and using unequal variance *t*-tests to estimate. The association between the SSOSH score in the baseline survey and the intervention group, as well as the probability of improvement in self-stigma of help-seeking (i.e., reduction in score from baseline to follow-up), were explained using a logistic regression model. The binary outcome (self-stigma of help-seeking improved or not improved) came from a Bernoulli distribution with the probability of improvement πimpr.

Based on the researchers’ prior work, the following scales (in addition to the Vogel scale) were used in the survey. All of our measures were established scales with literature referenced to verify their validity and reliability. We did not modify any scales: the Social Isolation Scale (SIS) [43]; reduced stress, leading to the adoption of the Perceived Stress Scale (PSS) [44]; and identification with characters, intended to be captured by the Perception of Narrative Performance Scale (PNPS) [45]. The Abbreviated Profile of Mood States—Adolescents Scale (POMS—A) [46] was adopted to measure any potential iatrogenic effects from the intervention, as well as to capture fluctuations in mood states from pre- to post-test to control for individual mood fluctuations. These prior studies indicated that the *Let’s Talk* suicide prevention videos (and plays) might achieve their effects by providing identifiable role models, sharing testimonials, describing narrative storylines with which the audience engaged, and by role-modeling open, frank discussions of suicide, stigma, suicidal ideation, and barriers to seeking professional help in a crisis.

#### 2.2.3. Data Analysis: Qualitative

To learn about connections between the numeric scores and the free responses, two experienced qualitative researchers (study authors) (one with a PhD in communication studies and one with a MS degree in psychology—both female) analyzed the components of the Vogel SSOSH scale (25 fixed-response questions) from all three studies. Only responses to three of the open-ended questions were analyzed: (1) “How did the video make you feel?”; (2) “Did the video change your attitude towards mental health problems? If so, how?”; and (3) “Did the video influence your attitudes towards professional counseling? If so, how?” Responses had no upper word limit, with instructions to provide as much or little detail as wished and to skip any questions that did not apply.

A comparative analysis was conducted for each open-ended question in relation to respondents’ numeric score changes. The team used grounded theory, a well-known methodology employed in research studies to discover or construct theory from data, systematically obtained and analyzed using comparative analysis [47] and textual analysis, a method used by researchers to describe and interpret the characteristics of a recorded or visual message [48], to enable us to draw inferences from the rich comments that the datasets provided.

Initially, respondents were categorized into rank order according to their quantitative changes in SSOSH scores, by the value of change, from greatest reduction to largest increases. The researchers then proceeded with a more in-depth data-driven analysis of the qualitative content and how respondents described their experiences across all the questions by performing independent analyses. As data analysis progressed, we conducted reflective, comparative team discussions that included consistency reviews to enhance validity, following COREQ (Consolidated criteria for Reporting Qualitative research) guidelines [49].

## 3. Results

### 3.1. Participant Characteristics

Data were from a total of 371 students over three studies (Table 1 and Table 2). Study 1 had 29 participants who completed both surveys. Originally, 87 college students were contacted, 40 enrolled, and 29 participants ultimately completed the study—a 33% follow-up rate. Participants in Study 2 included 291 college students. Initially, 800 potential participants were contacted, 596 enrolled, and 291 participants ultimately remained at the end of the two-week study—a 49% follow-up rate. Study 3 had 51 participants who completed both baseline and follow-up surveys. Enrollment started at 78 college students—a 65% follow-up rate.

### 3.2. Quantitative Analysis

#### 3.2.1. Comparisons of Means

In Study 1, lower rates of self-stigma of seeking help (SSOSH) were observed among students in the shorter video group (7-min *Let’s Talk* video). For respondents in that group, we estimated the mean SSOSH score decrease to be 4.16 (SE = 1.67) more than the mean score decrease for individuals in the control group (*p* = 0.017).

In Study 2, greater mean changes in SSOSH scores were observed among students in the *Let’s Talk* video group and the *SOS* group compared to the control group, but only the *SOS* group had statistically significant evidence of that difference according to the two-tailed *t*-test. On average, the rates of reduction of SSOSH scores of the participants in the *SOS* group were 1.78 times greater than those in the control group (95% confidence interval: 0.01–3.54; two-sided *p*-value = 0.049). Meanwhile, the evidence of a larger mean reduction of SSOSH scores between participants in the *Let’s Talk* group and those in the control group was negligible.

In Study 3, linear models were fit to compare the SSOSH scores between the intervention and control groups while controlling for gender, age group, sexual orientation, ethnicity, and campus. There was no evidence that the mean SSOSH score differed between the intervention and control groups (*p* = 0.59).

There was no evidence that the students’ race, ethnicity, sexual orientation, age, or gender altered the impact of the intervention on any of the outcomes assessed in any of the three studies. On average across all three studies, SSOSH scores were reduced from the baseline survey to the follow-up survey, indicating improvement on average, though that summary is not a satisfactory description of the response in many individuals (Figure 2) and there is clearly valuable information in the individual heterogeneity of responses.

#### 3.2.2. Odds of Improvement

The odds of improvement (defined as a decrease of 3 units in SSOSH between baseline and follow-up at the individual level) in the *SOS* intervention group were estimated to be 2.3 times the odds of improvement in the control group after accounting for baseline SSOSH score, age, gender, sexual orientation, and ethnicity. The data provide highly suggestive evidence that increased odds of improvement were associated with the *SOS* video (the two-sided *p*-value = 0.03 from Wald’s test). However, the estimated increase of 1.67 times in the odds of improvement among participants in the *Let’s Talk* intervention group compared to those in the control group was small relative to its standard error, so the data were consistent with the equal odds of improvement among participants in both the *Let’s Talk* intervention and the control.

There was strong evidence that the SSOSH score of the baseline survey was related to the odds of improving self-stigma of help-seeking for both the *SOS* and *Let’s Talk* videos (the two-sided *p*-value of the z-test was <0.01). For the same sexual orientation, ethnicity, gender, and group, an increase of 10 units in SSOSH score in the baseline survey was associated with a 2.099 times increase in the odds of self-stigma of help-seeking improvement, with an associated 95% confidence interval from 1.32 to 3.33 (Figure 3).

### 3.3. Qualitative Themes

Responses to qualitative questions from Study 1 were combined with responses from Study 3. Self-reported changes in response to the video were compared with individual stigma score changes. There were three main findings unearthed about the *Let’s Talk* video that were not captured in the quantitative analysis: (1) large stigma score reductions (−4 to −10) were associated with two types of responses—a) strong reports of change and b) no report of change; (2) small changes in stigma scores (from 0 to −3) were associated with two types of responses—(a) strong reports of change and (b) reports of no meaningful reductions in stigma of professional counseling; and (3) increased scores (>0) were frequently associated with increased empathy towards victims and reduced faith in available help (Figure 4, Figure 5 and Figure 6). Because of the high number of respondents, Study 2 was not included in this portion of the study.

#### 3.3.1. Larger Stigma Reductions

Respondents with stigma score reductions of 4 or greater were divided into two groups (Figure 4). The comments associated with greater declines in stigma showed changes in behavioral intentions among at least half the respondents: “I’m actually reaching out for counseling as we speak for help with my anxiety. It’s important to ask for help when you need it”, said one white female, aged 25–34, with a stigma decline of 4; “Yes, it tells you that it doesn’t make you weak if you need someone to talk to”, said one white female, less than 18 years of age, with a stigma decline of 7; and “Yes, if I needed help for a mental issue I would seek it”, said another respondent, white female, aged 18–24, with a stigma reduction of 8.

Positive comments given in response to the question “Did the video change your attitude towards professional counseling?” were present among about half the respondents who had larger stigma score reductions: “Yes, it encouraged me to get help if I ever need it”, said a white female, aged 18–24, with a stigma score reduction of 5, and “Yes, because a lot of people don’t want to talk about it and it was a great way for kids to be aware about it”, said another white female, aged 18–24, with a stigma score decrease of 7.

However, the other half of respondents with score changes greater than 3 expressed no change in their attitudes towards professional counseling: “Not really”, “No, I already am aware of this and think it is beneficial”, and “No because I already go to a professional counselor, I have for about 5 years now and I find it extremely helpful” were responses given by white females ages 18–34 with stigma score reductions of 9–10 points.

In response to the question “Did the video change your attitude about mental health?” people with large stigma score decreases generally reported changes in the direction of increased empathy: “Yes, a lot of people struggle with mental health issues more than you may even know”, said a white female, aged 18–24, with a stigma score change of 9. Some individuals directly talked about how the video helped to normalize depression and suicidality, and it helped make people struggling with such emotions feel less alone: “It made me more aware of how common it is to feel that way and how it can happen to anyone at any time”, said a white female, aged 18–24, with a stigma score decrease of 8.

Similar to the results on attitudes towards counseling, some respondents with large decreases in stigma score reported that their attitudes towards mental health problems had not changed: “No, I already felt compassion for people with mental health problems BUT i think it would be helpful for a lot of people to watch this video”, said a 35–44-year-old white female with a score change of −8.

#### 3.3.2. Small Stigma Reductions

For individuals with minor SSOSH score deceases (less than 3 points change from pre to post), participants were again divided between those who reported no change and those who reported the video had changed their attitudes towards mental health problems and counseling, in the direction of more empathy and openness.

In response to the first question, one white female aged 45–54 with a stigma score change of −1, reported that “[The video] made me want to reach out and hug them all”. A younger white female, aged 18–25, said the video made her feel “Sad, I hope more people reach out to others that need help”. This strong expression of empathy towards the actors in the video indicates that the video was able to trigger a connection between viewers and characters, even among individuals with low changes in their stigma scores.

In response to the second question “Did the video change your attitude towards professional counseling? If so, how?”, students with low changes in stigma scores (e.g., −1) reported substantial shifts: “Going to a therapist isn’t bad, it makes you responsible”, said a white female, aged 18–25, with a reduction in SSOSH of 1. Another white female, aged 35–44 with a stigma score reduction of 2, said “It should be more readily available to everyone”. Another white female from the same age group said “Yes, especially realizing that getting help makes you stronger”. “Yes, I will seek counseling if needed”, said another female, also white aged 18–24, with a stigma score drop of 2.

When asked if the video changed their attitudes towards mental health problems, an 18–24-year-old white female, with a stigma score reduction of 2, replied, “Yes, to be more sensitive to people who experience suicide thoughts”.

As with the group of participants with large reductions, about half of the individuals who had small reductions in their stigma score maintained that their attitudes had not changed: “No, I feel that counselors, parents, friends should always encourage seeking professional counseling. Once I obtain my LPN, I will seek to employment at schools so that I can make a difference. We are all responsible in knowing and taking suicide seriously” said a white female, aged 45–54, with a score decrease of 1.

#### 3.3.3. Increased Stigma

One surprising result was that many individuals who experienced an increase in stigma score expressed a reduced stigma of suicide (Figure 6). Increased stigma scores were frequently accompanied by increased empathy towards victims of suicide—indicating that stigma of mental health problems was not always associated with changes in attitudes towards counseling. In response to the question “Did the video change your attitude about mental health problems?”, one white female, aged 25–34, said “Yes, it helped show the light more on the topic which makes it easier to talk about”. However, this same person had an increase of 1 in her SSOSH score and replied “No” in response to the question about whether the video had changed her attitude towards professional counseling.

Another white female, aged 18–24, with an increase of 3 in her SSOSH score, said “Yes [the video changed my attitude towards mental health]. I have seen it thru the eyes of someone going thru it or better yet fighting it”. That same individual also responded “No” to the question about counseling.

“It put a different thought perspective into why people commit suicide. I always thought it was an easy, selfish way out” said a white 18–24-year-old female whose stigma score increased by 5. In response to the question about professional counseling, this individual replied, “No, [the video] did not change my attitudes”.

Similarly, a 25–34-year-old white female, with a stigma score increase of 5, said “Yes” in response to the question about mental health problems; “My perspective on suicide is no longer that the person is selfish and a coward”. An older female student, aged 35–44 with a stigma score increase of 9, said that the video “made me think about the others around me. Maybe someone is having a difficult time and I could say something that could help”. Neither of these individuals reported increased openness towards professional counseling.

A common response to the video was grief: “[The video made me feel] Sad that there are people who feel that helpless and think suicide is the only option”, said a white female, aged 35–44 with a stigma decrease of 8. The sadness was expressed in empathy for victims of suicide both for the severity of their sense of hopelessness, as well as for their lack of access to people who might be able to help: “The video made me very sad that people feel so alone in this world. There are so many people here that care about you and I think that before you do this you should try to talk to them” said a white heterosexual female under age 18.

## 4. Discussion

Statistical tests based on assessing differences in means indicated statistical evidence for reductions in stigma for groups seeing a suicide prevention video compared to a non-intervention group (under the other assumptions associated with standard tests of significance). However, this approach aggregated individuals and did not attempt to directly assess whether the observed difference in average changes in score was practically meaningful relative to the uncertainty quantified in the statistical methods. For example, is a difference in average change of −5 on the survey instrument scale (SSOSH) enough to argue that one treatment intervention (a 17-min suicide prevention video shown online) tends to reduce stigma so much more than the other treatment (a 7-min suicide prevention video), such that one should be implemented over the other? Additionally, this would ignore how many individuals actually reduced their stigma by a practically meaningful amount (i.e., enough to increase the likelihood that they will seek professional help), which is really more important information to assess and to report to stakeholders. 

The average score reduction for a group from a survey meant to measure stigma reduction is hard to interpret unless the survey instrument scale is well-connected to practical outcomes and changes on the scale are well understood in terms of potential practical impact on individual decisions. Unless all people in a treatment group had very similar changes (very low variability in the change scores), the average leaves out important information about changes experienced by individuals and whether the changes are large enough to potentially change behavior. The rates of improvement (defined on an individual level) are unknown, and an observed average over a group of people can be associated with statistical evidence for a difference even if improvement rate is relatively low and vice-versa. Therefore, the first step in pursuing a more individual-focused, practically relevant, and useful analysis is to define a criterion for improvement (e.g., reduction in stigma) or decline (e.g., stigma increase) for an individual. To do this, we examined prior literature and qualitative comments to establish and justify a magnitude of change for an individual that is expected to be associated with a shift in attitudes to the extent that it would change behavior (e.g., stigma reduced enough that the person would actually seek help or encourage others to seek help).

Basing conclusions and recommendations about stigma-reducing interventions on summaries of scores for groups limits our ability to understand how and why the interventions may be affecting individuals differently. The present study was concerned with examining the potential effectiveness of stigma reduction interventions at the individual level. Our outcome variable of interest was self-reported intentions to seek help from formal services (measured by the SSOSH scale), specifically professional counseling (primary care or secondary/tertiary mental health services) and including talking therapy services. The term ‘help-seeking’ is used to denote all stages of the process, encompassing the initiation of, and engagement with, care. Our prior research [28] indicated that any decrease in stigma of help-seeking can increase an individual’s self-reported intentions to access professional help, a known preventive measure for suicide [39,40]. However, no literature is available to indicate how much stigma reduction is needed for an individual to experience change and how best to measure changes in individual-level stigma [40]. Based on an analysis of individual quotes, we stick by our initial belief that any amount of decrease in stigma of help-seeking can be meaningful for an individual. Even students with small reductions in SSOSH scores, e.g., 1, reported seemingly meaningful changes.

The fact that many of the psychology students in Study 3 (often nursing or health care students) who had increases in their SSOSH stigma scores reported decreased stigma of suicide or suicidal people indicates a few things. One is that perhaps the stigma of *suicide* is very different from the stigma of help-seeking; changing perceptions about both may be important to preventing suicide. Indeed, Batterham et al. [50] developed the Stigma of Suicide Scale to measure attitudes towards people who attempt or die from suicide. Preliminary validity tests showed that more than 25% of respondents think people who suicide are “weak”, “reckless”, or “selfish”. The SOSS scale was not included in this study due to our pilot findings that responses were not associated with exposure to the suicide prevention videos we studied. Clearly, the SSOSH scale does not capture other facets of stigma (e.g., stigma of suicide).

Future research should explore the relationship between the different types of stigma. For example, most of the individuals with SSOSH increases in our study (14 of 20) reported reduced stigma towards suicidal people. One concern is that the experience of increased grief over others’ suicidality or a personal memory of loss may inhibit a person from embracing the effectiveness of sources of professional help. In other words, if a person’s friend sought help and did not survive suicidal thoughts, that person may lose faith in the effectiveness of such help.

A second finding is that the SSOSH scale—or perhaps any quantified approximation of complex social science constructs (e.g., human attitudes and emotions)—is an imperfect measure and may reflect changes in attitudes among people differently. SSOSH scores in our study seemed to mean different things for each individual. Increased scores were sometimes associated with self-reports of reduced stigma of help-seeking. Because stigma is such a complicated and multi-faceted construct to measure, it would be surprising if a single instrument could be identified to work for individuals in all sorts of contexts with all sorts of backgrounds. The challenge of validity in social science is not a new idea, but researchers who evaluate public health interventions (e.g., suicide prevention materials) have yet to widely adopt mixed methods approaches that could help minimize some of the common validity errors (Drost, 2011).

Third, we found that baseline SSOSH scores predicted the odds of improvement. When we calculated the odds of individual improvement in a logistic regression model, after accounting for the baseline SSOSH score and the previously mentioned descriptive information about the participants, our results showed that a person is more likely to reduce their stigma of help-seeking the higher their baseline score.

Finally, our results showed that most of the intervention group respondents qualitatively expressed more positive attitudes towards help-seeking, even with low changes (e.g., −2) in scores. These data, although small in number, gave us confidence that small changes in stigma scores could be meaningful at the individual level. Based on the number of individuals with score changes in the −2 and −3 range who expressed substantial changes, we decided to use −3 as a benchmark for the SSOSH score change needed to indicate meaningful change at the individual level.

In terms of which video to use, we showed statistically significant results nationwide using linear models for the *SOS* video and seemingly meaningful changes using qualitative self-reports for all of the videos. Some of our results we redeemed inconclusive because we disseminated to Montana Psychology students late in the semester, so their levels of literacy regarding suicide and stigma were potentially higher than students on other campuses. Batterham et al. observed that “participants who were studying psychology or had a psychology degree tended to be less stigmatizing and were more likely to view suicide as an outcome of isolation or depression”. This finding suggests that people benefit from an educational background in psychology, in terms of reducing stigmatizing attitudes and increasing understanding of why people suicide. Though the topic of suicide is not routinely covered in the undergraduate psychology curriculum, a core understanding of depression and other psychological disorders may be sufficient to promote these attitudes [50]. However, our findings indicated that literacy is insufficient as a suicide prevention program, since empathy for victims was sometimes contradictory to the perceived efficacy of professional counseling.

The average score reduction from a scale designed to measure stigma is hard to interpret for multiple reasons: (1) the average scores do not necessarily indicate how individuals have responded, (2) instruments designed to measure stigma are not typically connected to practical outcomes based on potential behavior changes, and (3) it is often not clear what larger group of people the average participant in any sample may reasonably represent. The idea that there is an “average person” associated with an average change in score is not necessarily the case. It is not clear that mean scores should be the focus of analyses, particularly in contexts like mental health research. In other words, unless all people in an intervention group had very similar changes (low variability in change), knowing the average would leave out important information about changes experienced by individuals and whether the changes are large enough to potentially change behavior (i.e., practically meaningful).

The reliance on mean score changes obfuscates variations occurring among individuals in any group and make it difficult for researchers to determine how much change is necessary for individuals to experience meaningful effects from an intervention or stimulus. For example, our results indicated that a difference in average change of five points on survey instrument scale like (SSOSH) is not enough to argue that one treatment intervention (a 17-min suicide prevention video shown online) tends to reduce stigma so much more than the other (a 7-min suicide prevention video) that one should be implemented over the other. Using standard estimates of significance ignores how many individuals actually reduce their stigma by a practically meaningful amount (i.e., enough to increase the likelihood that they will seek professional help), which is really more important information to assess and to report to stakeholders.

### Sample Limitations

The decision to study college students was based on the belief that, if enrolled in college, participants would have some access to mental health services, along with increased suicide rates for young adults worldwide [51]. The intervention effect may have been diluted by self-selection and recruitment bias. For example, the group of participants in Study 3 showed pre-existing mental health awareness, and some of them were traditional and non-traditional students pursuing careers in psychology and health care. Not all participants started from the same baseline levels of stigma, and some students reported that the video did not change their attitudes because they were already aware of the importance of mental health care. “It did not change my mind in anyway because I believe in professional counseling as in the power of prayer. So, I do believe that professional counseling is a must for everyone”, said one white female nursing student, aged 18–24. “My attitude towards mental health problems has always been positive, because I am not one to judge if someone has depression”. This limitation leads us to recommend collecting more information about participants’ baseline mental health awareness and experience in future studies.

The fact that score decreases were not always associated with reports of positive change leads us to believe that the scale may serve as a useful measure of stigma of help-seeking in some individuals, but not others. Though no pattern was observed in terms of who the scale seemed to work for in terms of age and other demographics, we believe this is to be expected given the complexity of stigma as a construct and the challenges associated with measuring it.

Another limitation involves the gender bias of the samples in Studies 1 and 3, where 86–90% of participants were female. This was due to the fact that a vast majority of students at the state university where the study was conducted are female, and the imbalance is even greater with psychology classes. However, the fact that gender was not associated with outcomes in our study, even in Study 2 where males and females were evenly split, indicated that having a greater portion of females may not have impacted the results.

Unfortunately, we had unequal sample sizes in our largest study, Study 2, where more than half the sample of the second treatment group dropped out before completing the follow-up survey. This unexpected attrition rate may have been due to the lower favorability of the second treatment, the *SOS* video, or just chance. The result was that the *Let’s Talk* treatment group had almost double the number of participants completing the study.

Another limitation had to do with the different designs of each study. While in Study 1, follow-up data were collected at one-week post-test, in Study 2, they were not collected until two weeks after the intervention. This difference in timing may have impacted the outcome scores in a way that is not known. It would be good to replicate the studies with a consistent follow-up design and to evaluate the difference in outcome measures as they relate to time of data collection.

Furthermore, another limitation was the exclusive use of college students; there is a need for a similar study of adults to determine the effects of informational interventions on their stigma of help seeking regarding suicidal ideation.

## 5. Conclusions

Four main themes emerged: (1) small changes in stigma scores were associated with individual reports of meaningful reductions in their attitudes towards professional counseling; (2) increased empathy towards victims of suicide and other mental health problems sometimes indicated increased empathy for victims of suicide and decreased openness in professional help; (3) empathy towards victims sometimes took the form of increased scores and grief or sadness, possibly thwarting the potential for help-seeking; and (4) self-reports of decreased stigma were not always associated with increased openness towards help-seeking. Results also indicated that small stigma score charges, not meeting statistical significance, were often associated with increased openness towards seeking help. These findings shed light on the nuances of individual stigma-levels and responses, as they relate to standardized measures of stigma and indicate that interventions to reduce stigma associated with suicide and help-seeking need to be analyzed at the individual level.

We showed a decrease in beliefs about stigma from a group of recruited college students nationwide. We also demonstrated in our pilot a significant decrease of Montana college students’ beliefs about the stigma surrounding help-seeking for mental health services. We believe our approach may translate into more young adults’ attitudes and behavior changes towards seeking help.

It is important to realize that an observed average over a group of people can be associated with statistical evidence for an intervention effect even if improvement rate is relatively low and vice-versa. Therefore, the first step in pursuing a more individual-focused practically relevant analysis is to define criteria for practically meaningful changes at the level of an individual (improvement associated with a reduction in stigma or decline associated with an increase in stigma) [52].

While stigma can take individual or collective forms and can be directed at oneself or others, analyses focused on estimating population or group means using average scores from a group of people and a particular instrument have failed to identify individual differences in response to interventions, as well as differences in backgrounds and pre-intervention ideas. Going straight to group averages of scores from a particular instrument also assumes that all relevant information is captured in the number associated with the answers to the survey questions.

Defining the criteria for a practically meaningful change (or “improvement”) can lead to more rigorous and in-depth analyses focused on whether individuals improve [52]. A focus on individuals can lead to greater insight into the intervention, as well as the instrument chosen to attempt to measure stigma. Statistical analyses can still be carried out to estimate the odds of positive change (or rates of improvement), rather than only a default focus on assessing evidence for differences in mean scores (or changes in scores) over groups of individuals.

An approach starting from definitions of meaningful changes at the individual level opens the door to a more holistic and deep analysis of the data, including opportunities to better integrate qualitative data collected at the level of the individual to the analysis of data considered quantitative.

To do this, we examined qualitative comments to establish and justify a magnitude of change for an individual that is expected to be associated with a shift in attitudes to the extent that it would change behavior (e.g., stigma reduced enough that the person would actually seek help or encourage others to seek help). We are unaware of other studies that have taken this approach and believe that it has promise in terms of developing a methodology for examining individual responses.

## Figures and Tables

**Figure 1 ijerph-18-06180-f001:**
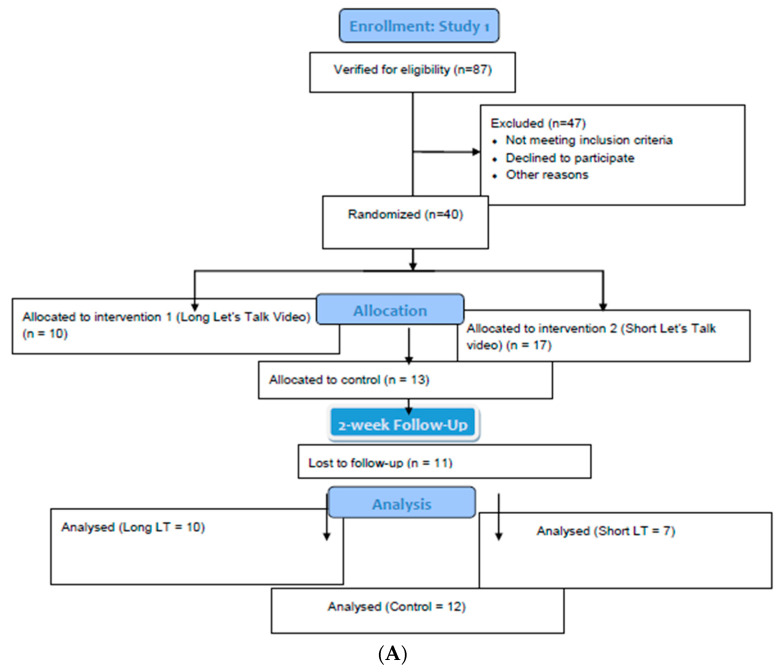
(**A**) Flow Diagram for Study 1; (**B**) Flow Diagram for Study 2; (**C**) Flow Diagram for Study 3.

**Figure 2 ijerph-18-06180-f002:**
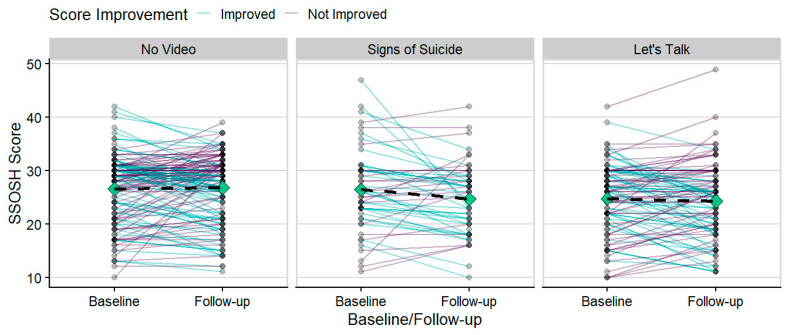
In Study 2, focusing only on the changes in the overall mean SSOSH scores in the intervention groups (dashed lines) ignores individual improvements (green lines) or declines (purple lines) in the perceived self-stigma of help-seeking before and after watching the videos. Note that the No Video control had similar variability among individuals, and the changes (good or bad) should not be necessary attributed to the videos.

**Figure 3 ijerph-18-06180-f003:**
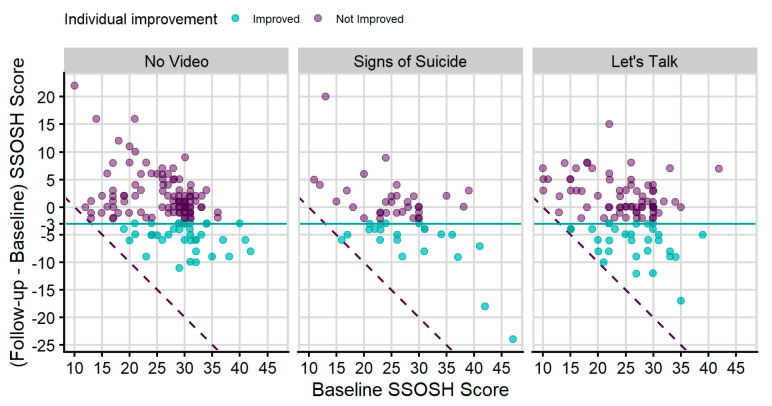
Plot of change in SSOSH against baseline SSOSH score for Study 2. The dashed line shows the greatest possible improvement for each baseline score. The horizontal line is the practically meaningful individual improvement cutoff at −3.

**Figure 4 ijerph-18-06180-f004:**
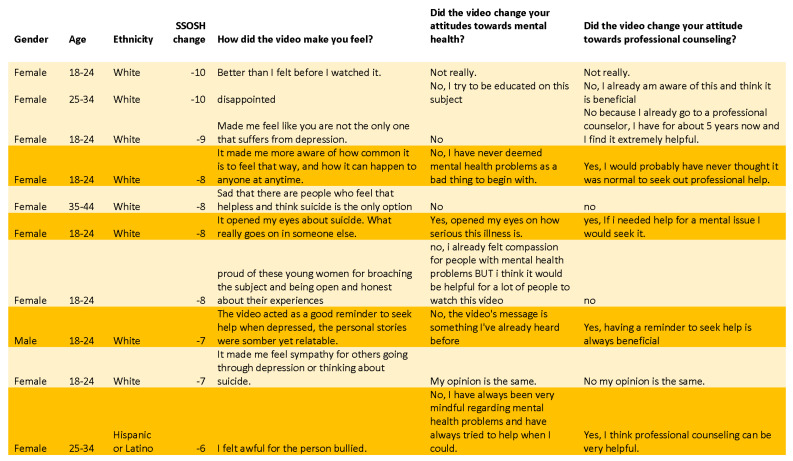
Group 1: large stigma reductions (respondents with stigma reductions > 3, i.e., a score change from −4 to −10). Participants were divided into two groups: those who reported increased openness to help-seeking (dark orange) and those who reported no change in attitudes towards counseling (light orange).

**Figure 5 ijerph-18-06180-f005:**
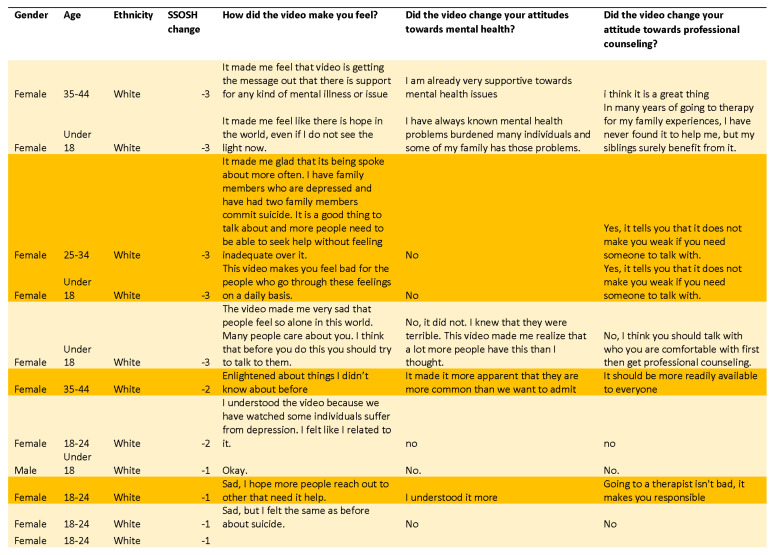
Group 2: small decreases in stigma score (0 to −3). Participants were divided into two groups: those who expressed minor increased openness to counseling (dark orange) and those who expressed no change in attitudes towards counseling (light orange).

**Figure 6 ijerph-18-06180-f006:**
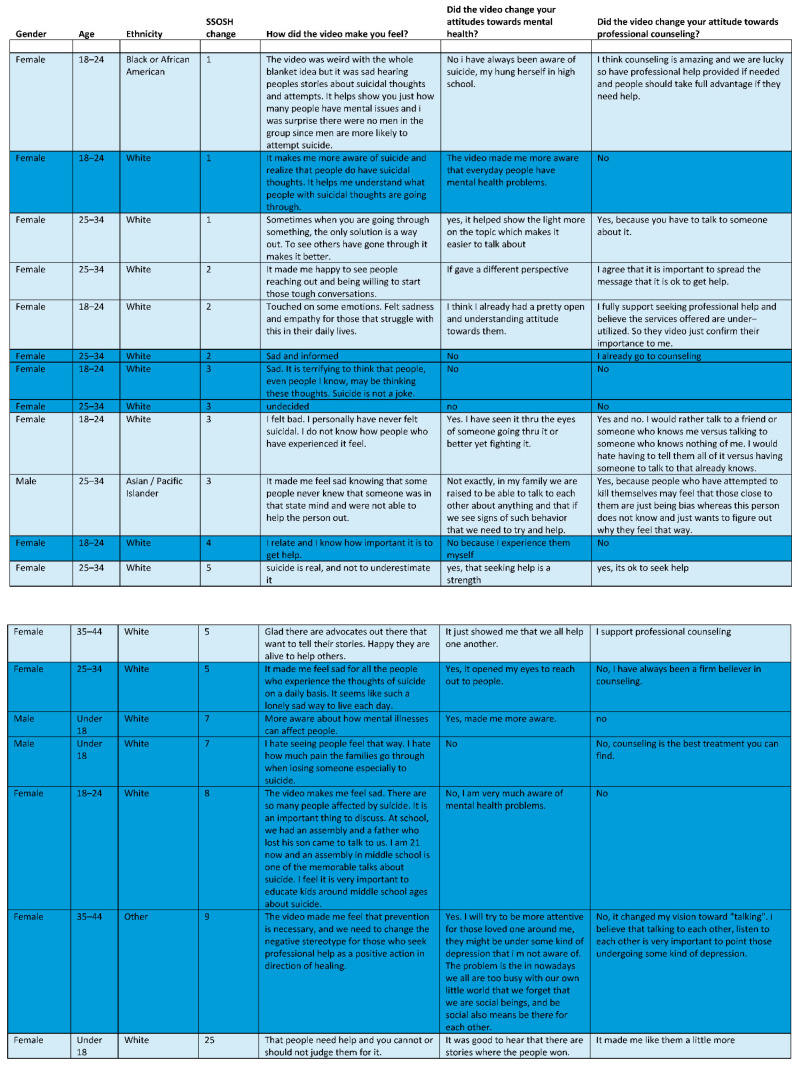
Group 3: increases in stigma score (>0). Participants were divided into two groups: Those who reported small increases in positive attitudes towards counseling (light blue) and those who reported no change (dark blue).

**Table 1 ijerph-18-06180-t001:** Enrollment for all studies.

Study	Sample Size at Final Data Collection	Group	Final Group Size
		Control	12 (41%)
Study 1 (includes Study 1A)	29	Long Let’s Talk	10 (34%)
		Short Let’s Talk	7 (24%)
		Control	135 (46%)
Study 2	291	Let’s Talk	102 (35%)
		Signs of Suicide	54 (19%)
Study 3	51	Let’s Talk	51 (100%)
TOTAL	371		

**Table 2 ijerph-18-06180-t002:** Demographics of participants in all three studies.

	**Categories**	**Study 1**	**Study 2**	**Study 3**
Age	18–25	17 (59%)		19 (37%)
	25–34		291 (100%)	16 (31%)
	35–44	7 (24%)		7 (14%)
	45–54	3 (10%)		2 (4%)
	55–64	1 (3%)		
	Under 18	1 (3%)		7 (14%)
	**Categories**	**Study 1**	**Study 2**	**Study 3**
Ethnicity	Asian/Pacific Islander		31 (11%)	2 (4%)
	Black or African American	1 (3%)	42 (14%)	1 (2%)
	Hispanic or Latino	1 (3%)	22 (8%)	2 (4%)
	Multiethnic		27 (9%)	
	Native American or American Indian	1 (3%)		1 (2%)
	Not Specified			1 (2%)
	Other			1 (2%)
	White	26 (90%)	169 (58%)	43 (84%)
	**Categories**	**Study 1**	**Study 2**	**Study 3**
Gender	Female	25 (86%)	170 (58%)	46 (90%)
	Male	4 (14%)	117 (40%)	5 (10%)
	Non-binary		4 (1%)	
	**Categories**	**Study 1**	**Study 2**	**Study 3**
Sexual Orientation	Bisexual	1 (3%)	31 (11%)	1 (2%)
	Gay/Lesbian/Homosexual		8 (3%)	1 (2%)
	Heterosexual/Straight	28 (97%)	245 (84%)	49 (96%)
	Not Specified		7 (2%)	

## Data Availability

Not applicable.

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
