# Peer review of "The Perceived Stigma Reduction Expressed by Young Adults in Response to Suicide Prevention Videos"

_ijerph, 2021, doi:10.3390/ijerph18126180_

Round 1

Reviewer 1 Report

This is a very lenghtly manuscript making the point that changes in scores of  various stigma measures, in particular on the Self-Stigma of Seeking Help Scale at group level after exposure to suicide prevention material, fail to show possible important correlations at individual level regarding attitudinal shifts and openess toward seeking mental health for oneself  and/or support to other.

Unfortunately this way of reasoningh is fatally flawed since, at individual level, these shifts may be due to thousands of other reasons at individual level not controlled for ( being tired, in a bad mood, having lost the car keys, etc) that have nothing to with the intervention, considering that in their study at group level the intervention was no better than placebo in outcomes.

I have made the point before but Authors seems to ignore this clear problem. Indeed in their line of thoughts they should report as well meaningful positive changes at individual level after the placebo condition which would just show that SSOSH scores changes would not be always a good predictor at individual level of more positive attitudes to people with mental health problems and suicide

There is a minor point.

Line 26. Suicidal ideation is now the second-leading cause of death 

Obviously ideation is not a cause of death.

"Suicidation" is used elsewhere in tha paper by the way and there is no reason to believe that it has any advantage to suicidal ideation. So it is should be best avoided since the nomenclature regarding self harm and suicide is already complex and confusing

Author Response

Dear Reviewer 1,

Thank you for the constructive feedback on our mixed methods analysis of suicide prevention videos. We have explained how we have addressed each concern below. We feel the reviewers’ suggestions have greatly improved the quality of our paper and now re-submitting to the International Journal of Environmental Research and Public Health.

Reviewer 1

Section

Response

This is a very lengthy manuscript making the point that changes in scores of  various stigma measures, in particular on the Self-Stigma of Seeking Help Scale at group level after exposure to suicide prevention material, fail to show possible important correlations at individual level regarding attitudinal shifts and openness toward seeking mental health for oneself  and/or support to other.

Full manuscript

Revisions were made throughout to reduce repetition and increase clarity. The introduction and methods were thoroughly re-written.

Unfortunately this way of reasoning is fatally flawed since, at individual level, these shifts may be due to thousands of other reasons at individual level not controlled for ( being tired, in a bad mood, having lost the car keys, etc.) that have nothing to with the intervention, considering that in their study at group level the intervention was no better than placebo in outcomes.

Full manuscript, lines 116-127, lines 431-434.

The purpose of the paper was to compare results with statistically significant effects to those without, in order to address the possibility that traditional tests of significance may not do an adequate job of capturing meaningful changes. Edits were made throughout to clarify this point.

We added a section in the Background, lines 116-127, to discuss the pitfalls of aggregate measures. While individual score changes may be associated with the intervening variables mentioned by the reviewer, these intervening variables also affect group score changes. This argument does not address the pitfalls of relying on aggregate scores.

We now clarify in lines 430-434 that the Abbreviated Moods Profile measure does indeed capture the fluctuations in mood states referred to by the reviewer: “The Abbreviated Profile of Mood States – Adolescents Scale (POMS-A) [46] was adopted to measure any potential iatrogenic effects of from the intervention, as well as capture fluctuations in mood states from pre- to post-test to control for  individual mood fluctuations that may account for individual responses.”

We also want to clarify that the interventions (both SOS and Let’s Talk) were statistically better than the controls in 2 of 3 studies at reducing SSOSH scores. Therefore, we think this study has merit in terms of illustrating an effective intervention, as well as raising red flags about how such approaches are evaluated.

Line 26. Suicidal ideation is now the second-leading cause of death

Line 66

This sentence has been modified to accurately attribute deaths to suicide, not ideation.

"Suicidation" is used elsewhere in the paper by the way and there is no reason to believe that it has any advantage to suicidal ideation. So it is should be best avoided since the nomenclature regarding self harm and suicide is already complex and confusing

Full manuscript

All references to “suicidation” have been removed.

Reviewer 2 Report

This manuscript presents the results of three related studies of group and individual perceptions of and changes in stigma among college students who have watched suicide prevention videos. The focus was on attitudes about suicide and particularly the seeking of mental health services for oneself or others. The study results identify that suicide prevention videos, as stigma reducing interventions, may have an influence on perceptions of stigma and that even small decreases might be meaningful in the case of stigma of help seeking.

The strengths of the manuscript include the use of experimental methods, with treatment groups and controls, as well as the utilization of instruments employed in previous research. Another strength is the mixed-methods design that included not only traditional experimental qualitative and group results, but additionally included individual qualitative results that provide insights into not simply an overall, aggregate result but also the variety of respondent results. The inclusion of a national sample of college student respondents is another strength.

There are some issues that might be addressed.

  • Studies: The combined descriptions of the individual studies in the method section provided piecemeal and mixed information that made it difficult to determine precisely how each individual study was conducted, from the standpoint of methods as well as participants and recruitment from one study to the next. Short but separate study descriptions would be easier to understand and, if desired, replicate.
  • Measures: In addition, there is no separate description of the measures employed, as these descriptions are mixed in with the already mixed study information. Are these measures reliable and valid (and sources)? Were the measures developed in relation to suicide stigma types, or stigma generally? Were there changes made in the measures questions if they were not focused already on suicide-related stigma? Or are these generic measures of stigma?
  • Videos: There is little information provided about the two videos employed other than their length and a citation associated with them. Why these videos? Where did these films come from? Are they available? Have they been used widely or at all in suicide prevention efforts? Are there features/themes etc. of these two videos that make them good candidates for this study of stigma of self-help? Any recommendations about the kind of information that others producing such intervention materials might include in their videos?
  • The Sample Limitations section addresses the potential gender bias of the samples, but there is no mention in Study 2's sample regarding why the number of participants in the Signs of Suicide video group was half that for the control and Let's Talk groups. As a national study, why the stark discrepancy? There were also smaller numbers in the Study 1 group as well in this cell (though in fairness, the numbers were small in all cells there). Is there a reason for this aspect?
  • In the section 2.1 Study Design and Participants, in the mixed descriptions of the studies, on lines 121-122 it is stated that "Post-test data were collected after only one week" (for Study 3 only?), but in the immediately following paragraph, lines 124-125, it states that "Data were collected immediately after the video and two weeks post-intervention…". This appears on the surface to perhaps be a contradiction, though it may well be an issue with slightly different methodologies associated with the 3 studies. As noted, separate descriptions of the studies would likely eliminate such uncertainty of methodological details. In a different aspect of the follow-up measures, whether retesting after 1 or 2 weeks, if still seems unknown whether any effects of the interventions had lasting impact on stigma of help-seeking. This issue is not raised in the Sample Limitations section (lines 459 ff). Another limitation is the use exclusively of college students; there is a need for similar study of adults to determine the effects of informational interventions on their stigma of help seeking regarding suicidal ideation.
  • A small point, but the term "suicidation" is employed in the "1. Introduction" paragraph. This term, from Cohen's 1969 article, is used in the present context to refer to what seems like exclusively, "suicide ideation" (as denoted by the parenthetical "suicidal ideation" after the term first appears) of "suicidal thoughts." This term is not a familiar one nor is it used widely in the suicidology literature. In fact, that term does not even appear in the Annual Review of Psychology article (reference 3, Joiner et al., 2005) that is cited at the end of the sentence in which the term appears and is used. Those familiar with the literature on this topic may well be unnecessarily distracted by this unfamiliar terminology as used here. Its use (rather than simply "suicidal ideation") does not seem to be important to the manuscript or its goals.
  • Another small point, immediately preceding the use of the term "suicidation," is a sentence that states that "Suicidal ideation is now the second-leading cause of death for young adults" (lines 27-28). Suicidal ideation is not a cause of death. Fatal outcomes of suicidal attempt, that is "suicide," is a cause of death and accurately, is the second leading cause for those aged 10-14, 15-24, and 25-34 in the most current (2019) USA mortality data.

Author Response

Reviewer 2

Thank you for the constructive feedback on our mixed methods analysis of suicide prevention videos. We have explained how we have addressed each concern below. We feel the reviewers’ suggestions have greatly improved the quality of our paper and now re-submitting to the International Journal of Environmental Research and Public Health.

The combined descriptions of the individual studies in the method section provided piecemeal and mixed information that made it difficult to determine precisely how each individual study was conducted, from the standpoint of methods as well as participants and recruitment from one study to the next. Short but separate study descriptions would be easier to understand and, if desired, replicate.

Methods, section 2.1.1, lines 233-297

Thank you for pointing out this unintentional confusion. We have now added short, distinct and detailed descriptions of each study in the methods section. We hope our flow chart will also help clarify what was done.

In addition, there is no separate description of the measures employed, as these descriptions are mixed in with the already mixed study information. Are these measures reliable and valid (and sources)? Were the measures developed in relation to suicide stigma types, or stigma generally? Were there changes made in the measures questions if they were not focused already on suicide-related stigma? Or are these generic measures of stigma?

Methods, lines 211-219; lines 222-223; lines 428-431

All of our measures were established scales with literature referenced to verify their validity and reliability. We did not modify any scales. In the Study Design section, we introduce our key outcome measure: “[14]. The Self-Stigma of Seeking Help scale [SSOSH] by Vogel, Wade, and Haake [28] was ultimately chosen was used as the basis for our aggregate analysis based on our pilot results coupled with the literature showing that reductions in estimated mean perceived self-stigma are tied to the increases in help-seeking behavior for psychological services [24,35–40].”

“We used the Extended Parallel Process Model (EPPM) [41,42](Witte 1992, 1994)  to examine the threat and efficacy components related to the goals of the Let’s Talk program [31]”

Later on in Methods, we introduce the other measures employed:

“…the Social Isolation Scale (SIS) [43]; reduced stress, leading to the adoption of the Perceived Stress Scale (PSS) [44]; and identification with characters, intended to be captured by the Perception of Narrative Performance Scale (PNPS) [45]. The Abbreviated Profile of Mood States – Adolescents Scale (POMS-A) [46]”

There is little information provided about the two videos employed other than their length and a citation associated with them. Why these videos? Where did these films come from? Are they available? Have they been used widely or at all in suicide prevention efforts? Are there features/themes etc. of these two videos that make them good candidates for this study of stigma of self-help? Any recommendations about the kind of information that others producing such intervention materials might include in their videos?

Lines 148-193

Thank you for pointing out this omission. We have now added a section in the Background to clarify the videos’ content, origin and availability, entitled Video as Intervention for Stigma Reduction. The themes of the videos include stories by adolescents related to their experiences with suicidal thoughts, alienation or loss due to suicide. We believe that role-modeling, testimonials and narrative approaches to education around this sensitive topic are key.

The Sample Limitations section addresses the potential gender bias of the samples, but there is no mention in Study 2's sample regarding why the number of participants in the Signs of Suicide video group was half that for the control and Let's Talk groups. As a national study, why the stark discrepancy? There were also smaller numbers in the Study 1 group as well in this cell (though in fairness, the numbers were small in all cells there). Is there a reason for this aspect?

Limitations, lines 781-785

Thank you for pointing this unfortunate limitation out. We have attempted to address it by adding the following lines to the Limitations section: “Unfortunately, we had unequal sample sizes in our largest study, Study 2, where more than half the sample of the second treatment group dropped out before completing the follow-up survey. This unexpected attrition rate may have been due to lower favorability of the second treatment, the SOS video, or just chance. The result was that the Let’s Talk treatment group had almost double the number of participants completing the study.”

In Study 1, relatively equal numbers were recruited for each condition, and completed all surveys. However, due to missing data for some subjects, data cleaning resulted in unequal sample sizes.

In the section 2.1 Study Design and Participants, in the mixed descriptions of the studies, on lines 121-122 it is stated that "Post-test data were collected after only one week" (for Study 3 only?), but in the immediately following paragraph, lines 124-125, it states that "Data were collected immediately after the video and two weeks post-intervention…". This appears on the surface to perhaps be a contradiction, though it may well be an issue with slightly different methodologies associated with the 3 studies. As noted, separate descriptions of the studies would likely eliminate such uncertainty of methodological details. In a different aspect of the follow-up measures, whether retesting after 1 or 2 weeks, if still seems unknown whether any effects of the interventions had lasting impact on stigma of help-seeking. This issue is not raised in the Sample Limitations section (lines 459 ff). Another limitation is the use exclusively of college students; there is a need for similar study of adults to determine the effects of informational interventions on their stigma of help seeking regarding suicidal ideation.

Section 2.1.1; Limitations lines 784-793

The description of methods used in each study have been clarified in Section 2.1.1.

The following limitations were added to the Limitations section:

“Another limitation has to do with the different designs of each study. While in Study 1, follow-up data was collected at 1-week post-test, in Study 2 it was not collected until two weeks after the intervention. This difference in timing may have impacted the outcome scores in a way that is not known. It would be good to replicate the studies with consistent follow-up design, and to evaluate the difference in outcome measures as they relate to time of data collection.

“Furthermore, another limitation is the use exclusively of college students; there is a need for similar study of adults to determine the effects of informational interventions on their stigma of help seeking regarding suicidal ideation.”

A small point, but the term "suicidation" is employed in the "1. Introduction" paragraph. This term, from Cohen's 1969 article, is used in the present context to refer to what seems like exclusively, "suicide ideation" (as denoted by the parenthetical "suicidal ideation" after the term first appears) of "suicidal thoughts." This term is not a familiar one…

Introduction

Use of the term suicidation has been eliminated from the text.

Suicidal ideation is not a cause of death.

Line 66

This error has been corrected.

Reviewer 3 Report

The article “The Perceived Stigma Reduction Expressed by Young Adults in Response to Suicide Prevention Videos” is quite interesting. However it requires some minor revisions, as described below.

Abstract

The present article stems from a limitation in the literature on stigma reduction when considering stigma against mental health or against suicide more specifically. In fact, according to the authors, aggregate measures for stigma reduction do not allow to understand how much stigma reduction is needed to determine a behavioral shift in seeking professional help for a mental health crisis. The aim of the present study is therefore to explore – both quantitatively and qualitatively - young adults’ perceived stigma reduction after viewing online-delivered suicide prevention videos.

The present section conforms with the journal guidelines in that it follows the style of structured abstracts, presenting all the relevant sections of the article.

Introduction

The present section presents in a clear and precise way the theoretical background behind this research. It starts by presenting general data on suicide and suicide prevention to then analyze stigma associated to suicide to finally move to a critique to aggregate measures for stigma reduction as they would not be able to capture the levels of stigma reduction needed to determine an attitudinal shift increasing the chances that a person would seek professional help. This latter introduces the aims of the present article which - besides trying to understand whether online-delivered documentaries for young adults could produce comparable levels to the SOS curricula or the Let’s talk live performances – tries to understand whether non statistically significant changes in stigma reduction could still be practically meaningful.

Minor changes are required for the present section. More specifically, at page 1 line 31, the expression “a plethora of suicide prevention efforts” could be expanded and a few examples of such interventions could be provided for readers. The same applies to page 1 line 32, where the authors could provide readers with the main factors contributing to suicidation. Moreover, the authors are invited to provide a short description of the Self-Stigma of Seeking Help scale (SSOSH).

Finally, the authors could consider including a relevant research article by Testoni and colleagues (2018) based on a Death Education course to prevent suicide among young people.

Testoni, I., Ronconi, L., Palazzo, L., Galgani, M., Stizzi, A., & Kirk, K. (2018). Psychodrama and Moviemaking in a death education course to work through a case of suicide among high school students in Italy. Frontiers in Psychology, 9. https://doi.org/10.3389/fpsyg.2018.00441

Materials and methods

Overall, the present section is well-organized, however, it should be further expanded and some aspects of it could be better clarified. More specifically, it could be useful for readers to clarify the reason why three different studies were implemented and why the authors chose to combine them in the present manuscript. Moreover, it could be useful for readers if the authors could also specify the timeline of the three studies. 

Study design and participants

The study design is described in a clear way and the ethical procedures behind the present research are explained in an extensive way. Nonetheless, the authors are invited to clarify whether or not a monetary reward was accorded to participants in exchange for their participation in the present research, since this element is only briefly mentioned in the text, but not explained in detail (p.3, lines 128-132).

Data analysis: quantitative

In the present section, the authors describe the quantitative measures used in this research. The authors should please describe the following four scales: “Social Isolation Scale (SIS)”, “Perceived Stress Scale (PSS)”, “Perception of Narrative Performance Scale (PNPS)”, “Abbreviated Profile of Mood States – _Adolescents Scale (POMS-A)”. As regards the above-mentioned scales, the authors should provide readers with basic information on, for instance, what they measure, the number of included items and the type of scoring. Moreover, they could clarify when the scales were used in the three studies. Finally, the authors are invited to re-phrase the expression “the messy topic of suicide” (page 4, line 174), in order to conform to the scientific language used in the rest of the manuscript.

Data analysis: qualitative

The present section describes the qualitative measures used in the present study. The authors are invited to describe both Grounded Theory and textual analysis methodologies (p.14, line 185), so that readers do not have to resort to the references. Moreover, the authors could clarify the reasons leading them to analyze data from the first and the third study together.

Results

Participant characteristics

In this section, participants’ characteristics are presented in a clear way. The reported tables are used in an appropriate way, as they allow a visual and more immediate understanding of the data.

Quantitative analysis

The present section is described in a clear and appropriate way and the visual outputs help readers to better understand the quantitative results.

Qualitative analysis

The present section offers an interesting comparison between the results obtained from the quantitative and qualitative analysis. The authors are invited to clarify the reasons behind their choice to combine data from the first and third study. The reported tables offer interesting visual support to the text, although the formatting of the tables makes it sometimes difficult to see them. Finally, two minor changes apply to the present paragraph. A typographical error is present at page 8, line 259, where “three were three” should probably be written as “there were three”. The expression “XX” at page 8, line 272, could be substituted with the appropriate measure of stigma reduction.

Discussion

In the present section, the authors offer a clear description of the main results, capturing their complexity and comparing them with relevant existing literature. Although this is not a mistake, the authors could consider leaving out direct quotes which are more appropriate in the results section only.

Sample limitations

The present section offers a very accurate description of the study limitations regarding the sample/participants involved in the present research. Moreover, the authors present a clear and transparent description of the study limitations also in the discussion section, where they describe the complexity and difficulty encountered in the interpretation of average score reductions from a scale designed to measure stigma.

Conclusions

This section adequately presents the main findings of the present study and their implications for future research and practice. In general, the study points to the importance of individualized interventions when focusing on stigma reductions of both suicide and help-seeking. 

References

Overall, the formatting of the references conforms with the journal guidelines. The authors should however revise the present section nonetheless, to make sure all references are formatted correctly, since there are a few possible mistakes (see, for instance, reference number 5 and number 42).

Overview

Overall, the study deals with a very interesting and internationally relevant topic and provides insight into the importance of integrating both quantitative and qualitative measures regarding stigma toward suicide and help-seeking in case of mental health crisis. The article is written in a clear and accurate way and only minor changes should be made. More specifically, the most substantial change the authors are invited to apply is to clarify the description of the three studies included in the present manuscript.

Author Response

Reviewer 3

Line 71, 72, 114-121

The phrase “plethora of suicide prevention efforts” has been removed. Given the focus of the study, we opted not to go into a review of such interventions.

Great point about the factors of suicide. Line 72 has now been revised to read: “Many factors contribute to suicidal ideation [8–11]; the most common contributors are social isolation and stigma [8–11].”

The SSOSH scale has now been defined on lines 114-121: “The Self-Stigma of Help-Seeking scale is designed to measure a person’s attitudes towards professional psychological help-seeking; what they perceive as others’ views of professional counseling; and how they view others who seek professional psychological help [28]. The scale uses an agree-disagree scale on 12 or 6 points to measure perceived public stigma; willingness to seek psychological help; attitudes towards seeking professional help; and self-stigma. In the proposed model, one’s self-stigma and attitudes towards seeking help mediate the relationship between perceived public stigma and willingness to seek help [28].”

Minor changes are required for the present section. More specifically, at page 1 line 31, the expression “a plethora of suicide prevention efforts” could be expanded and a few examples of such interventions could be provided for readers. The same applies to page 1 line 32, where the authors could provide readers with the main factors contributing to suicidation. Moreover, the authors are invited to provide a short description of the Self-Stigma of Seeking Help scale (SSOSH).

Finally, the authors could consider including a relevant research article by Testoni and colleagues (2018) based on a Death Education course to prevent suicide among young people.

Lines 108-112

This is a great suggestion, and we have added the Testoni et al. (2018) study to the Background section in lines 108-112.

The authors are invited to clarify whether or not a monetary reward was accorded to participants in exchange for their participation in the present research, since this element is only briefly mentioned in the text, but not explained in detail (p.3, lines 128-132).

Lines 306-310

Thank you for pointing out this omission. We have added the following text: “Monetary incentives were calculated at minimum wage, $8.15/hour, pro-rated for the estimated amount of time spent. Since two surveys, believed to last at least 20 minutes each, were combined with videos lasting from 7 to 17 minutes long, participants were anticipated to spend up to one hour on the study. Therefore, each participant was given $8.15 to participate, awarded upon completion of the follow-up survey.” 

The authors should please describe the following four scales: “Social Isolation Scale (SIS)”, “Perceived Stress Scale (PSS)”, “Perception of Narrative Performance Scale (PNPS)”, “Abbreviated Profile of Mood States – _Adolescents Scale (POMS-A)”. As regards the above-mentioned scales, the authors should provide readers with basic information on, for instance, what they measure, the number of included items and the type of scoring. Moreover, they could clarify when the scales were used in the three studies. Finally, the authors are invited to re-phrase the expression “the messy topic of suicide” (page 4, line 174), in order to conform to the scientific language used in the rest of the manuscript.

Lines 114-121; line 459

The suggestion to define and describe the scales is an excellent one. The only scale used as an outcome measure, SSOSH, has been properly defined in lines 114-121. Unfortunately, due to lack of time on the revision, we were unable to better define the other scales. We apologize for this omission.

The “messy topic of suicide” phrase has been changed to scientific language.

The authors are invited to clarify the reasons behind their choice to combine data from the first and third study. The reported tables offer interesting visual support to the text, although the formatting of the tables makes it sometimes difficult to see them

Line 465

Great catch. Indeed qualitative data from all three studies were analyzed. This error has been corrected in line 465.

We apologize for the difficulty with formatting. If the reviewer can clarify which tables or figures need improvement, we can re-format.

Typo: “Three were three…”

Line 548

The typo has been corrected.

Typo: XX

Line 561

The XX has been changed to a 7.

Although this is not a mistake, the authors could consider leaving out direct quotes which are more appropriate in the results section only.

Line 721-724

Direct quotes have been deleted from the Discussion.

The authors should however revise the present section nonetheless, to make sure all references are formatted correctly, since there are a few possible mistakes (see, for instance, reference number 5 and number 42).

References

The references have been re-formatted.

Round 2

Reviewer 1 Report

I have abuntantly made it clear in previous reviews the reasons why I find that is not good science  to analyse post hoc individual subjects scores shift of group design experiments. This, unfortunately, has not changed.

Author Response

Reviewer 1

Section

Response

This is a very lengthy manuscript making the point that changes in scores of  various stigma measures, in particular on the Self-Stigma of Seeking Help Scale at group level after exposure to suicide prevention material, fail to show possible important correlations at individual level regarding attitudinal shifts and openness toward seeking mental health for oneself  and/or support to other.

Full manuscript

Revisions were made throughout to reduce repetition and increase clarity. The introduction and methods were thoroughly re-written.

Unfortunately this way of reasoning is fatally flawed since, at individual level, these shifts may be due to thousands of other reasons at individual level not controlled for ( being tired, in a bad mood, having lost the car keys, etc.) that have nothing to with the intervention, considering that in their study at group level the intervention was no better than placebo in outcomes.

Full manuscript, lines 116-127, lines 431-434.

The purpose of the paper was to compare results with statistically significant effects to those without, in order to address the possibility that traditional tests of significance may not do an adequate job of capturing meaningful changes. Edits were made throughout to clarify this point.

We added a section in the Background, lines 116-127, to discuss the pitfalls of aggregate measures. While individual score changes may be associated with the intervening variables mentioned by the reviewer, these intervening variables also affect group score changes. This argument does not address the pitfalls of relying on aggregate scores.

We now clarify in lines 430-434 that the Abbreviated Moods Profile measure does indeed capture the fluctuations in mood states referred to by the reviewer: “The Abbreviated Profile of Mood States – Adolescents Scale (POMS-A) [46] was adopted to measure any potential iatrogenic effects of from the intervention, as well as capture fluctuations in mood states from pre- to post-test to control for  individual mood fluctuations that may account for individual responses.”

We also want to clarify that the interventions (both SOS and Let’s Talk) were statistically better than the controls in 2 of 3 studies at reducing SSOSH scores. Therefore, we think this study has merit in terms of illustrating an effective intervention, as well as raising red flags about how such approaches are evaluated.

Line 26. Suicidal ideation is now the second-leading cause of death

Line 66

This sentence has been modified to accurately attribute deaths to suicide, not ideation.

"Suicidation" is used elsewhere in the paper by the way and there is no reason to believe that it has any advantage to suicidal ideation. So it is should be best avoided since the nomenclature regarding self harm and suicide is already complex and confusing

Full manuscript

All references to “suicidation” have been removed.

Reviewer 2 Report

Review comments addressed.

Author Response

The combined descriptions of the individual studies in the method section provided piecemeal and mixed information that made it difficult to determine precisely how each individual study was conducted, from the standpoint of methods as well as participants and recruitment from one study to the next. Short but separate study descriptions would be easier to understand and, if desired, replicate.

Methods, section 2.1.1, lines 233-297

Thank you for pointing out this unintentional confusion. We have now added short, distinct and detailed descriptions of each study in the methods section. We hope our flow chart will also help clarify what was done.

In addition, there is no separate description of the measures employed, as these descriptions are mixed in with the already mixed study information. Are these measures reliable and valid (and sources)? Were the measures developed in relation to suicide stigma types, or stigma generally? Were there changes made in the measures questions if they were not focused already on suicide-related stigma? Or are these generic measures of stigma?

Methods, lines 211-219; lines 222-223; lines 428-431

All of our measures were established scales with literature referenced to verify their validity and reliability. We did not modify any scales. In the Study Design section, we introduce our key outcome measure: “[14]. The Self-Stigma of Seeking Help scale [SSOSH] by Vogel, Wade, and Haake [28] was ultimately chosen was used as the basis for our aggregate analysis based on our pilot results coupled with the literature showing that reductions in estimated mean perceived self-stigma are tied to the increases in help-seeking behavior for psychological services [24,35–40].”

“We used the Extended Parallel Process Model (EPPM) [41,42](Witte 1992, 1994)  to examine the threat and efficacy components related to the goals of the Let’s Talk program [31]”

Later on in Methods, we introduce the other measures employed:

“…the Social Isolation Scale (SIS) [43]; reduced stress, leading to the adoption of the Perceived Stress Scale (PSS) [44]; and identification with characters, intended to be captured by the Perception of Narrative Performance Scale (PNPS) [45]. The Abbreviated Profile of Mood States – Adolescents Scale (POMS-A) [46]”

There is little information provided about the two videos employed other than their length and a citation associated with them. Why these videos? Where did these films come from? Are they available? Have they been used widely or at all in suicide prevention efforts? Are there features/themes etc. of these two videos that make them good candidates for this study of stigma of self-help? Any recommendations about the kind of information that others producing such intervention materials might include in their videos?

Lines 148-193

Thank you for pointing out this omission. We have now added a section in the Background to clarify the videos’ content, origin and availability, entitled Video as Intervention for Stigma Reduction. The themes of the videos include stories by adolescents related to their experiences with suicidal thoughts, alienation or loss due to suicide. We believe that role-modeling, testimonials and narrative approaches to education around this sensitive topic are key.

The Sample Limitations section addresses the potential gender bias of the samples, but there is no mention in Study 2's sample regarding why the number of participants in the Signs of Suicide video group was half that for the control and Let's Talk groups. As a national study, why the stark discrepancy? There were also smaller numbers in the Study 1 group as well in this cell (though in fairness, the numbers were small in all cells there). Is there a reason for this aspect?

Limitations, lines 781-785

Thank you for pointing this unfortunate limitation out. We have attempted to address it by adding the following lines to the Limitations section: “Unfortunately, we had unequal sample sizes in our largest study, Study 2, where more than half the sample of the second treatment group dropped out before completing the follow-up survey. This unexpected attrition rate may have been due to lower favorability of the second treatment, the SOS video, or just chance. The result was that the Let’s Talk treatment group had almost double the number of participants completing the study.”

In Study 1, relatively equal numbers were recruited for each condition, and completed all surveys. However, due to missing data for some subjects, data cleaning resulted in unequal sample sizes.

In the section 2.1 Study Design and Participants, in the mixed descriptions of the studies, on lines 121-122 it is stated that "Post-test data were collected after only one week" (for Study 3 only?), but in the immediately following paragraph, lines 124-125, it states that "Data were collected immediately after the video and two weeks post-intervention…". This appears on the surface to perhaps be a contradiction, though it may well be an issue with slightly different methodologies associated with the 3 studies. As noted, separate descriptions of the studies would likely eliminate such uncertainty of methodological details. In a different aspect of the follow-up measures, whether retesting after 1 or 2 weeks, if still seems unknown whether any effects of the interventions had lasting impact on stigma of help-seeking. This issue is not raised in the Sample Limitations section (lines 459 ff). Another limitation is the use exclusively of college students; there is a need for similar study of adults to determine the effects of informational interventions on their stigma of help seeking regarding suicidal ideation.

Section 2.1.1; Limitations lines 784-793

The description of methods used in each study have been clarified in Section 2.1.1.

The following limitations were added to the Limitations section:

“Another limitation has to do with the different designs of each study. While in Study 1, follow-up data was collected at 1-week post-test, in Study 2 it was not collected until two weeks after the intervention. This difference in timing may have impacted the outcome scores in a way that is not known. It would be good to replicate the studies with consistent follow-up design, and to evaluate the difference in outcome measures as they relate to time of data collection.

“Furthermore, another limitation is the use exclusively of college students; there is a need for similar study of adults to determine the effects of informational interventions on their stigma of help seeking regarding suicidal ideation.”

A small point, but the term "suicidation" is employed in the "1. Introduction" paragraph. This term, from Cohen's 1969 article, is used in the present context to refer to what seems like exclusively, "suicide ideation" (as denoted by the parenthetical "suicidal ideation" after the term first appears) of "suicidal thoughts." This term is not a familiar one…

Introduction

Use of the term suicidation has been eliminated from the text.

Suicidal ideation is not a cause of death.

Line 66

This error has been corrected.